# Biodismantling, a Novel Application of Bioleaching in Recycling of Electronic Wastes

**Benjamin Monneron-Enaud** [1,*]**, Oliver Wiche** [2] 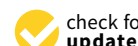 **and Michael Schlömann** [1,*]

1   Microbiology, Institute of Biosciences, TU Bergakademie Freiberg, Leipziger Straße 29,
    09599 Freiberg, Germany
2   Biology and Ecology, Institute of Biosciences, TU Bergakademie Freiberg, Leipziger Straße 29,
    09599 Freiberg, Germany; Oliver.Wiche@ioez.tu-freiberg.de
*   Correspondence: benjamin.monneron@ioez.tu-freiberg.de (B.M.-E.);
    Michael.Schloemann@ioez.tu-freiberg.de (M.S.); Tel.: +49-373-139-3011 (M.S.)

**Abstract:** Electronic components (EC) from waste electrical and electronic equipment (WEEE) such as resistors, capacitors, diodes and integrated circuits are a subassembly of printed circuit boards (PCB). They contain a variety of economically valuable elements e.g., tantalum, palladium, gold, and rare earth elements. However, until recently there has been no systematic dismantling and recycling of the EC to satisfy the demand for raw materials. A problem connected with the recycling of the EC is the removal of the components (dismantling) in order to recover the elements in later processing steps. The aim of the present study was to develop a new technique of dismantling using bioleaching technology to lower costs and environmental impact. In triplicate batch experiments, used PCBs were treated by bioleaching using an iron-oxidizing mixed culture largely dominated by *Acidithiobacillus ferrooxidans* strains supplemented with 20 mM ferrous iron sulfate at pH 1.8 and 30 °C for 20 days. Abiotic controls were treated by similar conditions in two different variations: 20 mM of $Fe^{2+}$ and 15 mM of $Fe^{3+}$. After 20 days, successful dismantling was obtained in both the bioleaching and the $Fe^{3+}$ control batch. The control with $Fe^{2+}$ did not show a significant effect. The bioleaching condition presented a lower rate of dismantling which can partially be explained by a constantly higher redox potential leading to a competition of solder leaching and copper leaching from the printed copper wires. The results showed that biodismantling—dismantling using bioleaching—is possible and can be a new unit operation of the recycling process to maximize the recovery of valuable metals from PCBs.

**Keywords:** bioleaching; e-waste; recycling; WEEE; PCB; biodismantling; components; rare earth elements; critical raw materials

## 1. Introduction

Electronic waste (e-waste) recycling currently is one of the biggest concerns regarding waste management. In 2017, Europe recycled 3.8 million tons of waste electrical and electronic equipment (WEEE), but, in the same year, 10.5 million tons of new electric equipment were put on the market [1]. In other words, Europe had a recycling balance of only 36% of WEEE in the objective of having a circular economy. Concomitantly, the European Union is concerned about "critical raw materials" which are materials presenting both a risk of supply and economic importance [2]. Many of these critical materials are chemical elements that have a supply risk due to a dependence on import; rare earth elements (REE), for example, are mainly supplied from China. Most electronic goods use a lot of those critical raw materials, e.g., REE in magnets of hard disk devices, yttrium and gallium in light-emitting diodes, tantalum in capacitors are among the best-known examples [3]. Therefore,

the stream of e-waste is a very good opportunity to relocate part of the production of those elements to Europe and treat the wastes at the same time.

Electronic waste has been proven to be a valuable material by its grade of precious and base metals, e.g., gold, palladium, silver, or copper. Gold constitutes the biggest economic interest with 50% of the possible revenue [4], but other metals are present also in significant amounts and still worth to be recovered. Valuable and critical metals are concentrated in the printed circuit boards (PCBs). These PCBs constitute, however, the most difficult part to recycle due to their high complexity of different combined materials in a broad range of concentrations. In fact, a few companies (e.g., Umicore) recycle PCBs to recover precious and base metals. Even if PCBs are recycled, REE are usually not recovered because of their very low concentrations, as explained by Umicore [5]. Additionally, only so-called high-grade PCBs can be economically processed.

The PCB is the "brain" of every device. It controls all accessories, e.g., screen, motor, battery, through decisions taken by the microcontroller or integrated circuit. The microcontroller is accompanied by diverse electronic components (EC) which aim at adapting electric signals to provide power and information to the different parts. Each of these EC is built based on the properties of specific metals. A majority of resistors uses ruthenium dioxide in the resistive paste for its high-temperature stability [6] and tantalum is present in capacitors for its dielectric properties [7]. Thus, a PCB includes the board itself and a multitude of ECs as subassemblies of the entire product, and specific metals are embedded in the components. This general construction constitutes the complexity of the PCB and leads to a dilemma. Recycling the PCB as a single piece is comparatively simple, but dilutes the dispersed elements. Separating the board and the components, in contrast, allows a more complete recycling, but also requires more energy and time. As industries aim at cost-efficient processes, the first approach is the one most widely used. Usually, PCBs are milled into powder with homogenous concentrations. The powder can easily be characterized and thus leads to comparable data. However, its main disadvantage is its high heterogeneity between the different batches and the low grade of the metals present when components are not first dismantled and separated. As reported by Ueberschaar [3], critical elements are detected in relatively high concentrations when components are characterized separately. As discussed later in this paper, this could lead to reconsidering the conclusion that the recovery of certain elements is impossible.

Dismantling is usually necessary to avoid pollution or to remove toxic compounds from the board such as the nonsolid electrolyte capacitors [8]. This is done either manually with pneumatic tools or by applying heat to melt the solder which is binding the components to the circuit [9]. This technique, like most pyrometallurgical techniques, suffers from low energy efficiency and produces toxic fumes which need to be treated. In all different cases, ECs have to be separated by affecting the solder which attaches the component to the board.

Solders are alloys which are required to conduct electricity, and to have a low melting point to fit the maximum temperature specified by the components. Originally, a tin-lead alloy known as Sn-37Pb was mainly used, thanks to its melting point at 183 °C under eutectic conditions. Because of environmental issues associated with the use of lead, solder technology has progressed towards lead-free solders. The most commonly used lead-free solder is SnAgCu also known as SAC [10]. The composition of SAC alloys includes different variations such as 96.5Sn-3.0Ag-0.5Cu (SAC305) or 95.5Sn-4.0Ag-0.5Cu (SAC405) [11].

Bioleaching provides a great opportunity to recycle e-waste, because (depending on the precise conditions) it may be able to handle the extraction of metals with lower costs than pyrometallurgical and conventional hydrometallurgical techniques [12]. Traditionally, bioleaching is applied for the extraction of metals from sulfidic ores as they occur in nature. From an engineering point of view, bioleaching is a ferric iron leaching in which the oxidant is recycled by ferrous iron-oxidizing microorganisms [13]. From an environmental point of view, it is one of the most efficient leaching processes because bioleaching can achieve the extraction of the metal at a moderate temperature, typically between 20 °C and 70 °C and with weak acid solutions, making the overall process a more ecofriendly way to

recover metals [14]. The main limitation of bioleaching is its lower reaction kinetic [14] which is often compensated by increasing the volume of the process. A good example of this is heap bioleaching of ores, during which the solution can run through the heap for weeks or months while still being economically feasible because of the thousands of tons of ores processed every day.

Research on bioleaching associated to PCB recycling has mostly focused on copper and gold recovery [15]. Copper can be extracted by biogenic ferric iron using iron-oxidizing bacteria. Gold extraction requires cyanogenic bacteria or fungi. Researchers are now focusing on upscaling and optimization of the extraction of copper to improve the rates and yield of bioleaching. The approach adopted by Hubau et al. is a two-step reactor to separate the production of biogenic ferric iron from the leaching reaction [16]. They achieved 96% recovery of Cu in a continuous 2.2 L reactor. While almost all studies used PCB powder or derivatives, Adhapure et al. showed that the extraction of copper from large pieces of PCB was also possible. Bioleaching of large pieces has the advantage of less mechanical pretreatment by avoiding comminution. However, since this study was done with depopulated PCBs, no conclusions on the behavior of the ECs could be drawn [17].

Another study reported on the effect of bioleaching solution on pieces of solder [18]. The authors described the production of ferric iron through biooxidation, followed by the treatment of the solder with the filtered solution. Bioleaching solution could successfully leach the SnAgCu solder, but not the SnPb one.

Based on the last two studies, the present work aims at proposing a new application for e-waste bioleaching. The ability to leach solder using bioleaching solutions, combined with the need to dismantle ECs leads to the idea of "biodismantling". Experiments are presented here to prove the concept of biodismantling using ferric iron produced by iron-oxidizing bacteria. In addition, this study presents a characterization of the component fractions which allows a new point of view on the "mineralogy" of waste PCB, thus opening opportunities to recover critical metals present in high concentration when initially enriched by dismantling and subsequent sorting.

## 2. Results

### 2.1. Dismantling

The progress of the corrosion of the PCBs under the different conditions was compared after 20 days (Figure 1). Bioleaching conditions and ferric iron control successfully removed ECs from the PCBs. Most of the dismantled components fell by themselves through the mixing action or were pushed out of the board with minimal force. The ferrous iron control, in contrast, showed very weak dismantling with only one component separated among the triplicate. The best condition of EC dismantling surprisingly was achieved by the ferric iron control and not by the bioleaching batches, reaching dismantling of 100% and up to 71%, respectively, relative to the total number of components (Figure 1c,d). Visual control showed that in the biochemical assays, some pins of components were missing and the visual appearance of components also changed more often in these batches.

The presence of bacteria as planned resulted in: the oxidation of the ferrous ions, the maximization of the concentration of the leaching agent and the stabilization of the oxidation-reduction potential (ORP) above 500 mV vs. Ag/AgCl throughout the experiment (Figure 2a). In comparison, the ferric iron control showed a decreasing trend of the ORP indicating the consumption of the oxidizing agent. At the same time, in the absence of sulfur compounds, the bacterial iron oxidation of $Fe^{2+}$ consumed protons resulting in a pH increase (Figure 2b). Both abiotic controls showed no significant change of pH.

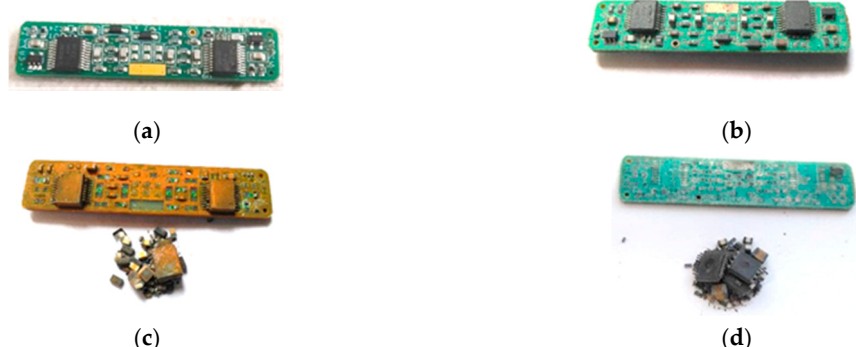

**Figure 1.** Photos of printed circuit boards (PCBs) resulting from different conditions after 20 days. (**a**) Original PCB without any pretreatment, (**b**) 20 mM ferrous iron leaching control, (**c**) biodismantling conditions with 10% inoculum and 20 mM ferrous iron, (**d**) 15 mM ferric iron leaching control.

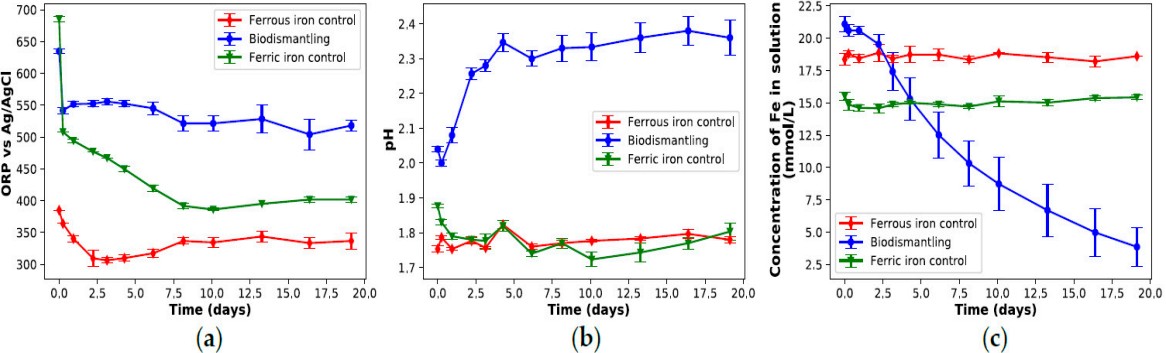

**Figure 2.** Evolution of the redox potential (**a**), pH (**b**) and iron concentration (**c**) during PCB biodismantling compared to ferric iron and ferrous iron controls. The first time point corresponds to oxidation-reduction potential (ORP) or pH, respectively, of the solution before the introduction of PCBs. Values are means from triplicates and error bars correspond to standard deviations.

Observation of the solutions showed clear differences in precipitation (Figure A2 in Appendix C). The biodismantling conditions showed an abundant brown precipitate which with respect to metal composition mainly contained iron and to a smaller extent tin and copper (Table 1). At the end of the 20 days, almost all the iron in the biodismantling solution was precipitated (Figure 2c). The precipitation of iron can be explained by its bacterial oxidation combined with an increase of the pH which lowered its solubility. The amount of precipitated copper was not significant compared to the total amount in the waste and consequently it did not play a role in the recovery of copper (Figures 3f and A1). Ferric iron controls showed a slight yellow/white precipitate, with respect to metals mainly composed of tin and to a smaller extent of iron and silver (Table 1). The relatively low redox potential of the ferric iron control suggests that the iron in the leaching process was largely reduced to ferrous iron. The reduced form of iron and the steady pH conditions thus prevented iron to precipitate. The ferrous iron control conditions did not show significant amounts of precipitation.

**Table 1.** Mass share of different elements in the precipitate on day 19. Percentages relative to the sum of analyzed metals (Fe, Ni, Cu, Zn, Ag, In, Sn) present in the precipitate.

| Sample | Time (day) | Mass Share (%) of Metal in Precipitate of | | | | | | | Sum (mg) |
|---|---|---|---|---|---|---|---|---|---|
| | | Fe | Ni | Cu | Zn | Ag | In | Sn | |
| Biodismantling | 19 | 89.3 | 0.1 | 2.8 | 0.4 | <0.1 | <0.1 | 7.4 | 181 |
| Ferric control | 19 | 8.4 | 0.1 | 0.5 | 0.9 | 1.4 | 0.3 | 88.4 | 72.5 |

Tin extraction reached 72% under the ferric control experiments while biodismantling and ferrous control setups reached only 17% and 0.4%, respectively (Figure 3). For both conditions containing ferric iron species, the total tin increased within the first hours after introduction of the PCB to 17% recovery. Obviously, this reaction could occur immediately after introduction of the waste PCB into the solution. The redox potential followed with a corresponding drop under both conditions. The passivation of tin is a possible explanation for the flat trend of the bioleaching condition. The Pourbaix diagram of tin indicates that a pH above 1 and a potential higher than 300 mV vs. SHE (standard hydrogen electrode) are in the passivation zone [19]. The increasing pH of the biotic condition coupled with the iron precipitation probably lead to a stable passivation layer of $SnO_2$. The ferric control, in contrast, benefitted from a lower potential, a lower and constant pH and possibly of the assistance of a galvanic effect between the copper circuits and the solder.

Copper was significantly extracted in the presence of bacteria (Figure 3), but no significant extraction of copper was observed under both control experiments. In the biodismantling setups constant and sufficiently high redox potentials may have overpassed a galvanic protection threshold while the controls could not. Additionally, observation of the gold-plated areas also showed gold detachment into the solution after a probable leaching of the copper support.

Finally, silver showed a significant increase in the precipitate of the ferric iron control reaching up to 20% recovery. Biodismantling and ferrous iron control did not show significant amounts of silver in the precipitate nor in the solution. The yield of silver was rather low compared to the tin recovery, although they were both constituents of the solder. Silver chloride may have been formed during the digestion of the precipitate sample by reacting with HCl contained in the *aqua regia*.

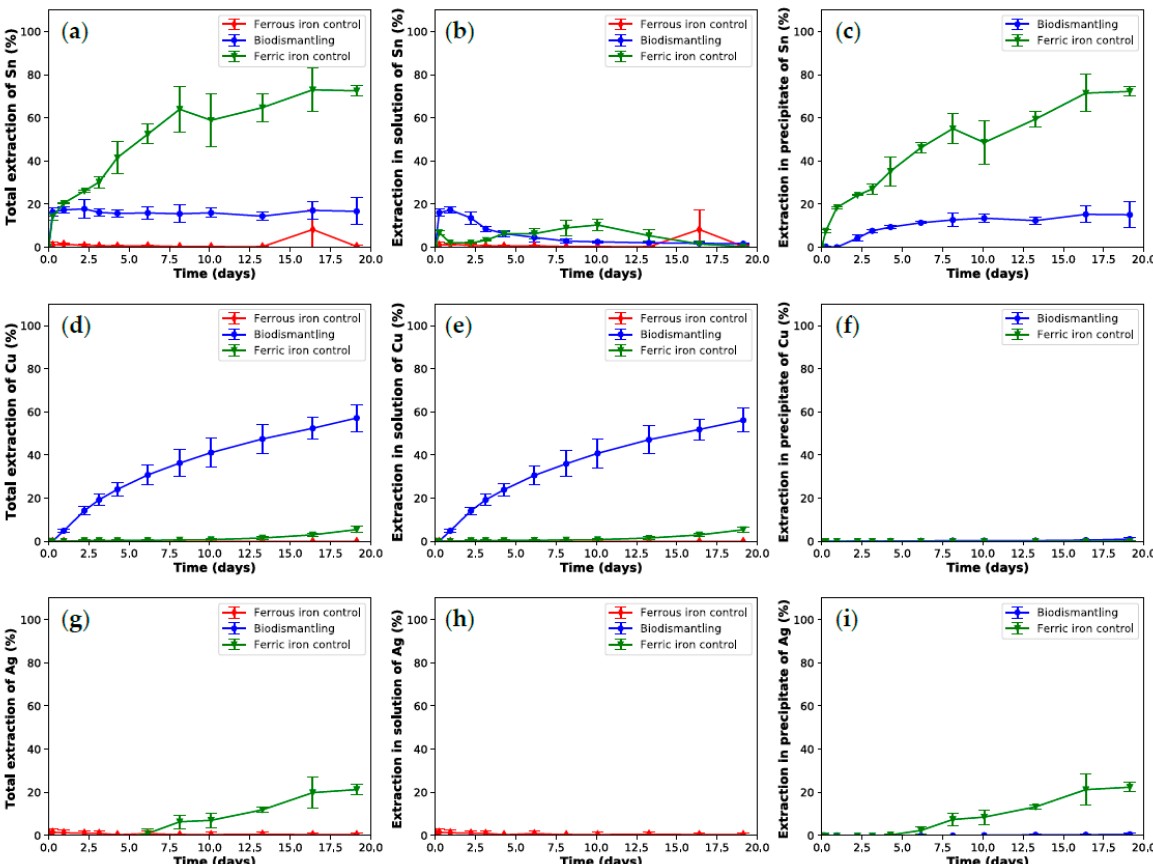

**Figure 3.** Extraction of Cu (**d**–**f**) in total and into solution as well as of Sn (**a**–**c**) and Ag (**g**–**i**) in total and largely into a precipitate, respectively, for biodismantling (blue circles), ferrous iron control (red rhombi), and ferric iron control (green triangles). Values are means of triplicates and error bars correspond to standard deviations.

## 2.2. Characterization of Component Fractions

Characterization of the components from the PCBs independently from the biodismantling experiment permitted the understanding of the "mineralogy" of the PCB (Table 2). As component fractions were considered on their own, high grades of critical metals were detected. The grade of the total waste was calculated based on the composition of the separated elements.

**Table 2.** Characterization of separated fractions of components after manual dismantling. The content of the total waste fraction was calculated from the contents in separated fractions weighted by their respective masses. Additional elements are shown in Appendix A.

| Element | Diode Like | Big Integrated Circuits | Small Integrated Circuits | White Capacitors-Like | Resistors-Like | Very Small EC | Brown Capapacitors-like | Depopulated PCB | Total Waste |
|---|---|---|---|---|---|---|---|---|---|
| | (wt.%) | (wt.%) | (wt.%) | (wt.%) | (wt.%) | (wt.%) | (wt.%) | (wt.%) | (wt.%) |
| Fe | 15.1 | 0.19 | 0.86 | <LD [1] | <LD | <LD | <LD | <LD | 0.17 |
| Ni | 10.3 | <LD | 0.55 | 7.76 | 3.91 | 8.72 | 2.99 | 0.55 | 1.07 |
| Cu | 0.81 | 8.62 | 36.0 | 6.00 | 1.61 | 6.43 | 4.07 | 15.3 | 12.9 |
| Zn | 1.51 | 0.03 | 0.31 | 0.15 | 0.32 | 0.50 | 0.11 | <LD | 0.04 |
| Sn | 3.25 | 1.03 | 1.50 | 2.01 | 6.39 | 8.64 | 0.57 | 2.79 | 2.36 |
| Ba | <LD | <LD | <LD | 0.12 | 0.01 | 35.9 | 11.1 | 0.03 | 1.16 |
| | (µg/g) | (µg/g) | (µg/g) | (µg/g) | (µg/g) | (µg/g) | (µg/g) | (µg/g) | (µg/g) |
| Mn | 1363 | <LD | <LD | 2450 | 192 | 586 | 672 | <LD | 149 |
| Co | 799 | 1 | 80 | 37 | 37 | 57 | 8 | <LD | 11 |
| Ga | 19 | <LD | <LD | <LD | <LD | <LD | <LD | <LD | <1 |
| As | <LD | <LD | <LD | <LD | <LD | <LD | <LD | <LD | <LD |
| Y | <LD | <LD | <LD | 13 | <1 | 24 | 714 | <LD | 57 |
| Ag | 365 | 35 | 211 | 380 | 228 | 729 | 41 | 1772 | 1226 |
| In | 107 | 32 | 48 | 65 | 206 | 278 | 18 | 86 | 73 |
| Sb | 15 | 2 | 7 | 4 | 10 | <LD | <LD | <LD | 1 |
| Dy | <LD | <LD | <LD | 1 | <LD | 9173 | 50 | <LD | 72 |
| Pb | <LD | 9 | <LD | <LD | 582 | <LD | <LD | <LD | 10 |
| Ru | <LD | <LD | <LD | <LD | 44 | <LD | <LD | NA [2] | 1 |
| Pd | 1 | 1 | 339 | 21 | 197 | 20 | 30 | NA | 12 |

[1] LD: limit of detection, calculated as three times the raw value of blank. [2] NA: data not available.

Depopulated PCBs were rather poor in critical metals, but hosted most of the tin, silver and copper because of the copper circuits and the solder residues. The diode fraction showed the presence of gallium suggesting the presence of GaAs semiconductors. Arsenic could not be properly quantified because of the higher background values. Ruthenium was detected in the transistor fraction which probably occurs as ruthenium oxide [6]. Barium was detected with up to 35 wt.% in the "very small components" fraction and with 11 wt.% in the brown capacitor fraction. It is probably a constituent of the capacitor ceramic as $BaTiO_3$. Up to 9000 µg/g of dysprosium and 700 µg/g of yttrium were found in those same fractions as Dy and Y can be used as dopants in the $BaTiO_3$ ceramic to improve its dielectric properties [20,21]. Indium was evenly distributed following the trend of tin, as each component may have some solder residues. This may be interpreted as an artefact of the mass spectrometry measurement because the naturally abundant $^{115}$Sn isotope (ca. 0.3 wt.%) can be detected as indium $^{115}$In.

## 3. Discussion

### 3.1. Biodismantling Concept

Biodismantling is the concept of beneficiation of waste PCBs by the separation of their components from the board using the activity of microorganisms.

The characterization of the separated component fractions showed high concentrations of critical metals such as Dy or Y. With an appropriate separation of the components after dismantling, an enrichment of these metals may be achieved by preserving the high grade of the fractions before recycling. Thus, biodismantling could help to make dismantling of the components a compulsory operation of recycling to maximize the recovery of the raw and critical materials.

The EC fractions have a minimized volume and mass compared to the original waste. In this study, the total fraction of components represented a third of the original PCB. Therefore, the separation of the components without any further sorting already would provide an enrichment by a factor of three for all EC materials. Additionally, the reduction of the mass and volume would be an advantage for handling the remaining waste.

The waste beneficiation process could further be enhanced by separating the components from each other. For example, the "very small components" fraction contained up to 9000 µg/g of dysprosium when considered on its own. In this fraction, which represented only 0.7 wt.% of the original PCB, the dysprosium content was increased 140 times as compared to the total PCB waste. The higher dysprosium content in the separated fraction is comparable to ore grade, and therefore its recovery should be economically feasible. Rare earth oxide ores in Chinese mines of Southern provinces, for example, contain 25,000 µg/g of dysprosium, while Bayan Obo (China, Inner Mongolia) holds 600 µg/g of Dy [22]. In contrast, the total amount of dysprosium in the total PCB waste accounted for only 70 µg/g, which is much less interesting to recover, even though this concentration is quite high for a circuit board. Priya and Hait reported concentrations of REE of about 1 µg/g for several elements in ground PCB from different appliances with the highest concentration of dysprosium reported as only 0.31 µg/g [23] which is below the value for the earth's crust (4.1 µg/g [24]).

To achieve the concentrations of the separated EC, a sorting process would be necessary and could be achieved according to different criteria. Integrated circuits have multiple pins for communication and are big enough to be discriminated from other ECs by size. Capacitors exist in different technologies, e.g., aluminum capacitors, multilayer ceramic capacitors (MLCC) or tantalum capacitors [25]. These differences in construction generate different shapes which should allow separation. Aluminum capacitors, for example, are cylinders while MLCC are cuboids. It would be important to separate them according to their technology to obtain the richest fractions possible. Resistors tend to be quite a homogeneous fraction, since for them the thick-film technology is dominant. Their content may vary according to the manufacturer, but ruthenium oxide is the most common resistive paste [6]. As the PCB used in our study did not contain any connectors or frames, the behavior of such components under biodismantling cannot be concluded from our experiment. These components are usually made of stainless steel which may interact with ferric iron depending on its quality.

Biodismantling combines two concepts which have already been studied separately, i.e., leaching of fully depopulated PCBs and leaching of solder. Thus, Adhapure et al. described the performance of bioleaching of large PCBs and showed successful leaching of copper using iron-oxidizing bacteria. However, to liberate the copper from its protective layer and to allow its reaction with ferric iron, a chemical treatment was needed [17]. Solder bioleaching has been studied by Hocheng et al. [18] by using iron-oxidizing and citric-acid-producing bacteria to dissolve solder. Biogenic ferric iron solutions could dissolve the Sn-Ag-Cu solder, but not the Sn-Pb one, while citric acid could dissolve both. The lead solder in presence of sulfate may have been passivated by the formation of poorly soluble $PbSO_4$ [10]. Therefore, biodismantling in the case of Sn-Pb solders may have to use citric-acid-producing bacteria to overcome the passivation problem.

In our study, experiments were conducted to prove the feasibility of biodismantling and to understand its mechanisms. Flask-scale leaching successfully dismantled ECs using ferric iron as an oxidizing agent. The ferrous iron control, in contrast, showed a complete absence of dismantling.

At first sight, two different mechanisms seemed to take place. Biodismantling was superior over the abiotic controls for copper leaching, whereas tin was preferably leached in ferric iron control set-ups. However, in both cases it is the $Fe^{3+}$ in acid solution which attacks the respective metals. The difference between the two set-ups is in the ORP. Copper oxidation requires a higher ORP than tin. When this condition is reached by the bacterial activity, leaching leads to the dissolution of copper circuits and removal of the components, by removing the support and pins of components. Selective leaching of tin, on the contrary, seemed to occur at lower ORP, i.e., when iron in the absence of bacteria

was not reoxidized. The difference in nobility allowed tin to be dissolved, but copper not. Therefore, only the solder dissolved and components were liberated. Further investigation of the influence of redox potential and galvanic effects are required. A system that would allow lower redox with the presence of bacteria may be selective as the ferric control conditions, and still be more efficient thanks to microbial activity. A two-step reactor system—decoupling iron production and leaching—or a continuous reactor system may be more suitable to achieve such redox control.

Additionally, it should be mentioned that bioleaching under our conditions ended with almost no iron in the solution because of the Fe(III) precipitation promoted by the relatively high iron concentration and the pH increase. A pH-regulated system (possibly together with lower total iron concentration) could then probably keep iron in solution until dismantling is complete.

### 3.2. Recovery of Metal Released during Biodismantling

During biodismantling, four different kinds of products are produced: the depopulated board, the separated ECs, the leaching solution and the precipitate. These elements should be separated, so that each can receive a specific treatment.

The board will still contain copper, if circuits have not yet been leached. For example, in so-called "multilayer PCBs" copper circuits are embedded in the resin support. Liberation is then necessary by comminution, pyrolysis or deconstruction of the resin. As the circuit boards also host flame retardant chemicals, they need to be handled with care to minimize the propagation of hazardous compounds [26]. The liberated copper can then be extracted by an additional bioleaching process, which, for obtaining the copper, would be followed by solvent extraction. The resin can be treated in different ways to remove toxic compounds and produce, e.g., fuel, new chemicals or filling material for composite material [27].

Obtaining metals from ECs as mentioned earlier would benefit from sorting before comminution and metal extraction. Processes certainly will vary with the type of EC and the targeted element. Whether or not there are additional chances to make use of bioleaching remains to be investigated.

The biodismantling solution contains solubilized metals, such as iron and copper. From this solution the copper can be won by solvent extraction, while the iron-containing acid solution can be recycled to the process.

The precipitate, depending on the degree of redox control, will contain Fe(III) minerals and also the tin and the silver from the solder (Table 1). At the pH used iron(III) would probably precipitate as jarosite by reacting with sulfate, hydroxide ions and a cation such as $K^+$, $Ag^+$, $Na^+$, $H^+$ etc. [28]. As mentioned above, one probably would try to minimize iron precipitation by redox and pH control as well as by possibly reducing the iron concentration. For tin and silver, one could see the precipitation as an opportunity to achieve easy purification. According to the Pourbaix diagram of tin [19] one may expect $SnO_2$ as solid phase at high redox potential and silver may have precipitated as chloride (AgCl). However, detailed analyses will have to show whether or not these assumptions are correct and how possibly the formation of such precipitates can be further optimized or used for separation. For example, increasing the chloride concentration may bring the tin, but not the silver, into solution and thus would make a separation of these elements possible.

Associated with the precipitate also the plated gold foils were found, which had been released into the solution by the action of the leaching agent. Gold cannot be dissolved by the ferric iron attack, so when gold is plated on copper circuits, galvanic effects may dissolve the copper below it and release the foil. Gold particles may sink down and be mixed with the precipitate.

## 4. Materials and Methods

### 4.1. Printed Circuit Boards (PCBs)

A batch of identical PCBs was obtained from an online scrap seller. The product was described as "Medical PCB", but the exact function of these PCBs is unknown. The PCBs were double-face mounted

components and had multilayers of copper circuits. Each PCB contained 77 components of various sizes, all as surface-mounted devices. Two zones were plated with gold, a rectangular area and a via, i.e., a stretch going all the way through the board.

### 4.2. Characterization of the Circuit Boards and Components

The printed circuit boards and the electronic components were characterized separately. Three representative PCBs were selected and were desoldered with a soldering iron. The free ECs were assorted in seven different categories based on visual criteria (Table 3) i.e., diode-like, big integrated circuits, small integrated circuits, white capacitor-like, brown capacitor-like, resistor-like and very small components. Each EC fraction was digested by a microwave-assisted aqua regia procedure, exposing the sample in a Teflon tube to 1.4 mL of an aqueous solution of 24% HCl and 14% $HNO_3$ at 200 °C for 30 min (including 15 min to reach the temperature). While the aqua regia procedure is a standard procedure in our lab and many others, it could not be applied to the depopulated PCBs due to their size. Depopulated PCBs were digested in two steps process at room temperature. First PCBs were soaked in 32.5% $HNO_3$ for 24 h. Then the remaining solid part and precipitates were leached with 18.5% HCl for 24 h. All digestion solutions were analyzed by ICP-MS (Xseries 2, Thermo Scientific) with 10 µg/L rhodium and rhenium as internal standards.

**Table 3.** List of obtained fractions after manual desoldering for characterization.

| Fraction Name | Mass of Fraction. Mean and Standard Deviation (mg) | Part of the Total Waste (wt.%) |
|---|---|---|
| Depopulated PCB | 2511 ± 5 | 67.3 |
| Diode-like | 30 ± 0 | 0.8 |
| Big integrated circuits | 638 ± 2 | 17.1 |
| Small integrated circuits | 61 ± 1 | 1.6 |
| White capacitor-like | 119 ± 5 | 3.2 |
| Brown capacitor-like | 295 ± 2 | 7.9 |
| Resistor-like | 61 ± 3 | 1.6 |
| Very small ECs [1] | 27 ± 2 | 0.7 |
| Total | 3733 ± 14 | 100 |

[1] The small size of the components made identification difficult. Therefore, it was decided to make a separated category.

### 4.3. Strains and Growth Conditions

The inoculum was prepared from a mixed culture designated BiCoNi4 as described elsewhere [29], which had been subcultured on $FeSO_4$. It consisted mainly (ca. 99%) of *Acidithiobacillus ferrooxidans* strains (unpublished data). The culture was grown on modified DSMZ medium 882 consisting of 132 mg/L $(NH_4)_2 SO_4$, 53 mg/L $MgCl_2 \cdot 6\ H_2O$, 27 mg/L $KH_2PO_4$, 147 mg/L $CaCl_2 \cdot 2\ H_2O$, and 1 mL/L trace element solution (62 mg/L $MnCl_2 \cdot 2\ H_2O$; 68 mg/L $ZnCl_2$; 64 mg/L $CoCl_2 \cdot 6\ H_2O$; 31 mg/L $H_3BO_3$; 10 mg/L $Na_2MoO_4$; 67 mg/L $CuCl_2 \cdot 2\ H_2O$). The medium was modified with respect to a pH of 1.8 and an $Fe^{2+}$ concentration of 20 mM as described previously [29].

### 4.4. Dismantling Experiment

Preliminary tests had indicated that bioleaching conditions needed pretreatment to liberate copper circuits from their protective coating (unpublished). Therefore, the PCBs were pretreated by soaking them in 10 M NaOH at room temperature for three days to remove the coating covering the copper circuits [17]. They were rinsed and brushed with deionized water, then rinsed with 70% ethanol and air-dried before use.

Flasks containing 200 mL medium were inoculated with 10 vol.% of culture containing $10^7$ cells/mL and incubated for two days at 30 °C before insertion of the pretreated (none sterile) PCBs. All experiments were done in triplicates. Two abiotic control conditions were run. A first one contained

20 mM ferrous sulfate but no inoculum, and a second one contained 15 mM ferric ions (also as sulfate). Flasks were placed for 20 days in a 30 °C shaking incubator at 140 rpm. Samples of 1.5 mL liquid, containing also formed precipitates, initially were taken every day, from day four every second day, and from day ten to twenty every third day. The precipitate and the supernatant were separated by centrifugation for three minutes at 20,000× *g*. Supernatant samples were stored after the addition of 65% nitric acid to a final concentration of 0.7% to increase the stability of ions. Precipitate samples were digested with and stored in 500 μL of aqua regia (23% HCl + 22% $HNO_3$) in an Eppendorf tube at room temperature for seven days until complete dissolution was achieved.

*4.5. Chemical Measurements*

pH and redox potential (vs. Ag/AgCl) were measured with Hamilton Polilyte Plus electrodes. All metal concentrations were analyzed by inductively coupled plasma mass spectrometry ICP-MS (Thermo Scientific-XSeries). Fe, Cu, Sn and Ag, were determined after sample preparation including dilution by a factor of 5000 with ultrapure water (<0.055 μS/cm) using the addition of 10 μg/L Rh and Re as internal standards.

Metal recoveries were calculated based on the values of the waste PCB characterization according to Equations (1) and (2) for metal concentration in the solution.

$$Y_{solution} = \frac{[M]_{solution}}{[M]_{max}}, \tag{1}$$

$$[M]_{max} = \frac{M_{waste} \cdot m_{waste}}{V_{flask}}, \tag{2}$$

where $Y_{solution}$ is the percentage of yield related to the metals in the solution, $[M]_{solution}$ is the concentration of the metal in the solution, $[M]_{max}$ is the maximum concentration corresponding to the complete dissolution of the waste, $M_{waste}$ is the massic fraction of metal in the waste PCB, $m_{waste}$ is the mass of waste in the flask, $V_{flask}$ is the volume of the reaction during the experiment.

Precipitate recovery values were calculated analogously in Equation (3). Apparent precipitate concentration was calculated according to Equation (4) to estimate the precipitate content in the reaction flask. Precipitate particles were assumed evenly distributed during the sampling to obtain a representative sample. The ratio of the precipitate over the solution was then considered equivalent in the sample and in the flask.

$$Y_{precipitate} = \frac{[M]_{precipitate}}{[M]_{max}}, \tag{3}$$

$$[M]_{precipitate} = \frac{[M]_{digestion} \cdot V_{digestion}}{V_{sample}}, \tag{4}$$

where $Y_{precipitate}$ is the percentage of yield related to the metals in the precipitate, $[M]_{precipitate}$ is the apparent concentration of the precipitate in the flask, $[M]_{digestion}$ is the concentration of the precipitate digestion solution which was measured by ICPMS. $V_{digestion}$ is the volume the precipitate digestion solution, $V_{sample}$ is the volume of sample taken from the flask.

## 5. Conclusions

Biodismantling is a new application of bioleaching in the recycling process of electronic waste. Its main application is to enhance the concentration of critical and precious materials imbedded in the electronic components. By separating the components from the board, a relatively high grade is preserved instead of being lowered by the grinding process. Considering sufficient sorting of the components after separation, some rare earth elements may become economically available by reaching a grade comparable to commercial ores. A concentration of 9000 μg/g of dysprosium has been measured in one of these separated fractions.

Experiments proved the feasibility of the concept and resulted in two different outcomes depending on the redox potential. High redox potential lead to leaching of copper circuits while lower redox potential preferentially leached solder.

Future work will be carried out to prove functionality in a more technical setting by designing an adapted reactor. Further investigations are also necessary to increase the solid load and the reaction rates. The pretreatment to remove the protective coating is an additional step and consequently generates additional costs, therefore studies are necessary to understand its role and possibly to avoid this step.

The benefits of dismantling are independent of biodismantling, but the low-cost production of ferric iron by bacterial activity may allow a more systematic dismantling and enhance the recovery of critical metals. The process for electronic waste treatment will be a combination of different technologies, where biodismantling could be a cost-effective first step with low cost and low energy consumption.

**Author Contributions:** Experiment, B.M.-E.; formal analysis, O.W.; writing—review and editing, B.M.-E., M.S., O.W. All authors have read and agreed to the published version of the manuscript.

**Funding:** The authors thank the Erich-Krüger Foundation for generous financial support.

**Acknowledgments:** The authors would like to thank Cristian Jorquera and Beate Erler for their TRFLP analysis on the BiCoNi4 culture, Javiera Norambuena Morales and Fabian Giebner for their support and advice during the initial stages of these studies and Sabrina Hedrich for valuable discussions and comments on the manuscript.

**Conflicts of Interest:** The authors declare no conflict of interest. The funders had no role in the design of the study; in the collection, analyses, or interpretation of data; in the writing of the manuscript, or in the decision to publish the results.

## Appendix A

**Table A1.** Characterization of separated fractions of components after manual dismantling. The content of the total waste fraction was calculated from contents in separated fractions weighted by their respective masses.

| Element | Diode-Like | Big Integrated Circuits | Small Integrated Circuits | White-Capacitors-Like | Resistors-Like | Very Small EC | Brown-Capacitors-Like | Depopulated PCB | Total Waste |
|---|---|---|---|---|---|---|---|---|---|
| | (wt.%) | (wt.%) | (wt.%) | (wt.%) | (wt.%) | (wt.%) | (wt.%) | (wt.%) | (wt.%) |
| S | <LD [1] | <LD | <LD | <LD | <LD | <LD | <LD | <LD | <LD |
| Fe | 15.10 | 0.19 | 0.86 | <LD | <LD | <LD | <LD | <LD | 0.17 |
| Ni | 10.32 | <LD | 0.55 | 7.76 | 3.91 | 8.72 | 2.99 | 0.55 | 1.07 |
| Cu | 0.81 | 8.62 | 35.96 | 6.00 | 1.61 | 6.43 | 4.07 | 15.26 | 12.88 |
| Zn | 1.51 | 0.03 | 0.31 | 0.15 | 0.32 | 0.50 | 0.11 | <LD | 0.04 |
| Sn | 3.25 | 1.03 | 1.50 | 2.01 | 6.39 | 8.64 | 0.57 | 2.79 | 2.36 |
| Ba | <LD | <LD | <LD | 0.12 | 0.01 | 35.89 | 11.14 | 0.03 | 1.16 |
| | (µg/g) | (µg/g) | (µg/g) | (µg/g) | (µg/g) | (µg/g) | (µg/g) | (µg/g) | (µg/g) |
| Mn | 1363 | <LD | <LD | 2450 | 192 | 586 | 672 | <LD | 149 |
| Co | 799 | 1 | 80 | 37 | 37 | 57 | 8 | <LD | 11 |
| Ga | 19 | <LD | <LD | <LD | <LD | <LD | <LD | <LD | <1 |
| As | <LD | <LD | <LD | <LD | <LD | <LD | <LD | <LD | <LD |
| Br | <LD | <LD | <LD | <LD | <LD | <LD | <LD | <LD | <LD |
| Y | <LD | <LD | <LD | 13 | <1 | 24 | 714 | <LD | 57 |
| Ag | 365 | 35 | 211 | 380 | 228 | 729 | 41 | 1772 | 1226 |
| Cd | <LD | <LD | <LD | 7 | <LD | <LD | <LD | <LD | 0 |
| In | 107 | 32 | 48 | 65 | 206 | 278 | 18 | 86 | 73 |
| Sb | 15 | 2 | 7 | 4 | 10 | <LD | <LD | <LD | 1 |
| Ce | <LD | <LD | <LD | 3 | <LD | <LD | 2 | <LD | 5 |
| Pr | <LD | <LD | <LD | <1 | <LD | <LD | <1 | <LD | <1 |
| Nd | <LD | <LD | <LD | 1 | <LD | <LD | <1 | <LD | <1 |
| Eu | <1 | <LD | <LD | <1 | <LD | 43 | 14 | <LD | 1 |
| Gd | <LD | <LD | <LD | <1 | <LD | 2 | 39 | <LD | 3 |

**Table A1.** *Cont.*

| Element | Diode-Like | Big Integrated Circuits | Small Integrated Circuits | White-Capacitors-Like | Resistors-Like | Very Small EC | Brown-Capacitors-Like | Depopulated PCB | Total Waste |
|---------|------------|-------------------------|---------------------------|-----------------------|----------------|---------------|------------------------|-----------------|-------------|
| | (µg/g) | (µg/g) | (µg/g) | (µg/g) | (µg/g) | (µg/g) | (µg/g) | (µg/g) | (µg/g) |
| Dy | <LD | <LD | <LD | 1 | <LD | 9173 | 50 | <LD | 72 |
| Pb | <LD | 9 | <LD | <LD | 582 | <LD | <LD | <LD | 10 |
| Ru | <LD | <LD | <LD | <LD | 44 | <LD | <LD | NA [2] | 1 |
| Pd | 1 | 1 | 339 | 21 | 197 | 20 | 30 | NA | 12 |
| Os | 1 | <1 | <LD | <LD | <LD | <LD | <LD | NA | 0 |
| Ir | <LD | <LD | <LD | 1 | <LD | <LD | <LD | NA | 0 |
| Pt | <LD | <LD | <LD | 4 | <LD | <LD | <1 | NA | 0 |

[1] LD: limit of detection, calculated as three times the raw value of the blank. [2] NA: data not available.

## Appendix B

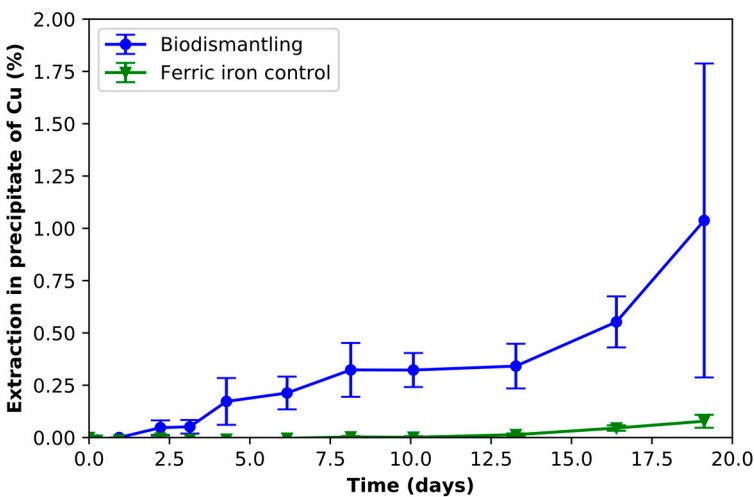

**Figure A1.** Extraction of Cu as precipitate for biodismantling (blue circles) and ferric iron control (green triangles). It shows the same data as Figure 3f, but here the Y axis scale is from 0% to 2% to have a better evaluation of the trend. Values are means of triplicates and error bars correspond to standard deviations.

## Appendix C

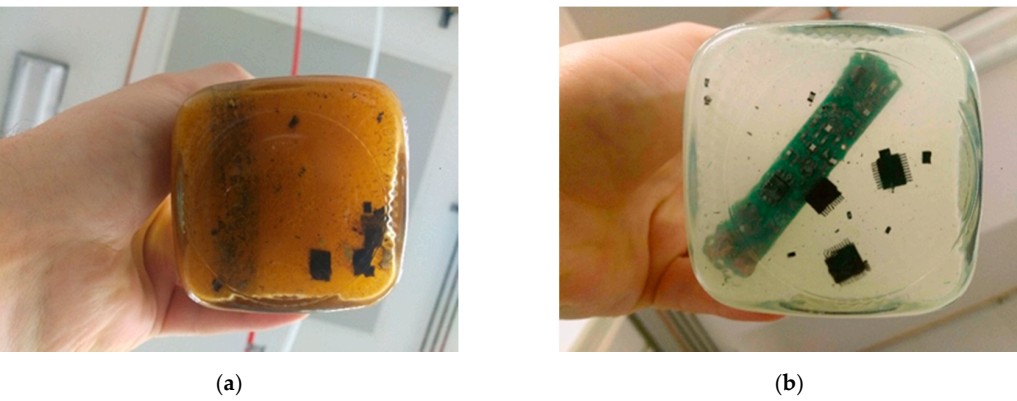

(**a**)                                                     (**b**)

**Figure A2.** Pictures of the aspect of the solution and the precipitate in the flask during biodismantling (**a**) and ferric chemical control (**b**).

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
