# Peer review of "Biodismantling, a Novel Application of Bioleaching in Recycling of Electronic Wastes"

_recycling, doi:10.3390/recycling5030022_

Round 1

Reviewer 1 Report

This is an interesting manuscript, with very useful information on the feasibility of separating components from a PCB , which is a major task to overcome in precious metal recycling from PCB. It also mentions the characterisation of different types of components mounted on boards while highlighting the challenging slow rate of process and therefore the limits for industrial application.

Overall, English language and style could benefit from proof reading by native English speaker. For example the term of "upgrading" repeatedly used in the discussion is confusing. Does the author means refining, or partially purifying?

Introduction:

The introduction is generally instructive, but some information mentioned could be more details; for example lines 37-39 a CRM is more than just a chemical element with a supply risk. Some details relevant to the work could also be added in the introduction: previous work tackling the same issue of solder removal with bioleaching is briefly mentioned in the discussion, however reviewing the state of the art in the introduction part is also essential as it helps the reader to understand the novelty of the present work.

Some details about the composition of the different solders should be provided.

Methods:

The digestion protocol mentions a microwave assisted digestion with Aqua Regia.. More details should be provided.

The bacterial strains are mentioned as a mixture.. can detailed composition  be provided?

The PCB and separated components were digested differently but no clear reason why?

A pre-treatment with NaOH was carried out, but was there any treatment without pre-treatment, which is not mentioned?

What’s the choice and significance of the 20 days time for the bioleaching? Was a time course carried out and 20 days chosen as optimal time?

Results

Lines 166 to 174 need formatting, line 171 onwards is the same text as the figure heading used below line 179...

A brown precipitate is mentioned in line 197; was the presence of Jarosite detected? See line 307 could maybe add some more discussion?

Please clarify how the extraction values reported in Fig. 3 are calculated.

Please modify the caption of Fig 3. to mentioned which metals shown on graph did separate in the solution, and which metals were present in the precipitate;

Mentioning the weight/amount of precipitate in each process, maybe a photo of the precipitates and their characterisation with analytical method, would improve the quality of this paper

Please clarify in the text more details about the physical separation of the components after the bioleaching: were the components totally separated from the board, or was it necessary to manually facilitate this separation?

With ferric control the amount of ferric ions (15mM) will be consumed  quickly... Is there enough ferric ion available to attack copper?  Can you estimate quantity of metals going into test in mM and compare this with quantity of Fe(III) present? Experiment at higher ferric ion concentration?

Can you estimate how many turnovers of ferrous to ferric occurs in bioleaching test?

Does the presence of metal species in the inoculum have an effect on bioleaching mechanism and apparent selectivity towards copper. Chloride ion in particular could have an effect...

Will tin not react with Fe3+ to form Sn4+ species which reduce Fe2+ to Fe?

Discussion:

The author mention 2 different mechanism, but it is not clear whether they’re proposing bioleaching process as an option to the ferric process, or simply an alternative for selectivity?

The discussion on recovery of metals would benefit from more references illustrating methods combined to bioleaching for recuperation. 

Reviewer 2 Report

Please see comments in the attached file.
